# Development of New Experimental Dental Enamel Resin Infiltrants—Synthesis and Characterization

**DOI:** 10.3390/ma15030803

**Published:** 2022-01-21

**Authors:** Doina Prodan, Marioara Moldovan, Andrea Maria Chisnoiu, Codruța Saroși, Stanca Cuc, Miuța Filip, Georgiana Florentina Gheorghe, Radu Marcel Chisnoiu, Gabriel Furtos, Ileana Cojocaru, Ada Gabriela Delean, Sanda Ileana Cimpean

**Affiliations:** 1“Raluca Ripan” Institute for Research in Chemistry, Babes-Bolyai University, 30 Fantanele Street, 400294 Cluj-Napoca, Romania; doina.prodan@ubbcluj.ro (D.P.); marioara.moldovan@ubbcluj.ro (M.M.); codruta.sarosi@gmail.com (C.S.); stanca.boboia@ubbcluj.ro (S.C.); miuta.filip@ubbcluj.ro (M.F.); 2Department of Prosthodontics, “Iuliu Hațieganu” University of Medicine and Pharmacy, 32 Clinicilor Street, 400006 Cluj-Napoca, Romania; maria.chisnoiu@umfcluj.ro; 3Faculty of Dental Medicine, Carol Davila University of Medicine and Pharmacy, 17–23 Calea Plevnei, 010232 Bucharest, Romania; georgiana.gheorghe@umfcd.ro; 4Department of Cariology, Endodontics and Oral Pathology, “Iuliu Hațieganu” University of Medicine and Pharmacy, 33 Moților Street, 400001 Cluj-Napoca, Romania; ada.delean@umfcluj.ro (A.G.D.); sanda.cimpean@umfcluj.ro (S.I.C.); 5Department of Horticulture and Food Science, Faculty of Horticulture, University of Craiova, 13 Al. I. Cuza Street, 200585 Craiova, Romania; i_cojocaru2005@yahoo.com

**Keywords:** dental resin infiltrants, white spot, degree of conversion, fluoride release, flexural strength

## Abstract

The aim of the present study was to obtain experimental infiltration materials, intended for the treatment of dental white spots, and to investigate them. Two series of infiltrants (P1–P6)/(P1F–P6F) were obtained, based on different monomer mixtures, without/with glass filler (with fluoride release ability). Each infiltrant from the second series contained the same amount of glass powder, and each infiltrant from the (P–PF) group contained the same resin composition. The characteristics of the experimental infiltrants were investigated by degree of conversion (DC), mechanical strength, water sorption (WS), and fluoride release, in addition to residual monomer for (P1F–P6F) infiltrants. The results were compared with those obtained for commercial Icon infiltrant. For the experimental infiltrants, without/with filler, the recorded DC was in the range of 58.27–89.70%/60.62–89.99%, compared with Icon (46.94%) 24 h after polymerization. The release of fluoride depends on the permeability of the polymer matrix, with respect to the water sorption, which may help to diffuse ions in the storage medium but which can also influence the release of residual monomers. The highest flexural strengths were recorded for the (TEGDMA/HEMA/Bis-GMA) infiltrants (133.94 ± 16.389 MPa/146.31 ± 7.032 MPa). The best experimental infiltrants were P2 and P2F (Bis-GMA/HEMA/TEGDMA).

## 1. Introduction

The demineralization of tooth enamel is followed by the appearance of white spots. These white spots appear due to the organic acids produced by bacteria. Therefore, there are losses of calcium ions, leading to an increase in enamel porosity and, subsequently, to the colonization of bacteria. To prevent the development of early enamel lesions, commonly used treatments are fluoride therapy, remineralization, casein–phosphopeptide–amorphous pastes, calcium phosphate, and laser therapy. However, fluoride treatments are limited to the surface of the lesion, not being able to reach the demineralized tissue. Thus, for the treatment of white spots, infiltrants have been used as a micro-invasive treatment option [1].

Icon (DMG America, Ridgefield Park, NJ, USA) infiltrant was introduced on the market only a few years ago, and, since then, there have been numerous studies demonstrating both the qualities and the limitations of the effectiveness of this material. With a low viscosity, the infiltration resin manages to fill the dental pores through a capillary phenomenon, strengthening the demineralized teeth and improving the resistance to the acid attack in the oral environment [2,3].

The main component of the Icon is triethylene glycol dimethacrylate (TEGDMA). The addition of bisphenol A glycidyl methacrylate (Bis-GMA) to experimental infiltrants, which has a more rigid molecular structure due to the aromatic group, aims to increase the flexural strength (FS) and the elasticity module—Young’s modulus (YM), as well as reduce the polymerization shrinkage [4]. TEGDMA is a hydrophilic monomer with a low molecular weight that can be more easily eluted after polymerization, in the aqueous medium, due to unreacted molecules, leading to adverse effects [5,6].

According to the literature data, the methacrylate-based composites contain C=C double bonds that remain unpolymerized in the polymer network and up to 10% unpolymerized monomers prone to elution and water sorption [7]. Furthermore, the water sorption (WS) leads to a decrease in the mechanical properties due to swelling and matrix plasticization [8].

The degree of conversion (DC) is closely related to the quality of the polymerization. The measurement is performed before (on the monomer mixture) and after the end of the curing process, at several points on the samples surface [9]. The quality of the three-dimensional polymer network, formed after polymerization, depends on the DC, which in turn can provide predictability in terms of the mechanical strength of the material [10], while the sorption and solubility can provide valuable information on the release of potentially toxic components that can promote cytotoxicity. If the DC is high, then the amount of monomer converted to the polymer is higher, reducing the possibility of residual monomers being released into the oral environment.

The novelty of this study is the obtention of two series of experimental infiltration materials, intended for the treatment of dental white spots, and their investigation in terms of the characteristics associated with this type of material, by comparing them with Icon Smooth Surface, one of the most used commercial infiltrants. Icon is an infiltrant based on the TEGDMA matrix. The present study also demonstrates the influence of other monomers in the resin matrix, in addition to TEGDMA, and/or of a solvent on some physicochemical and mechanical properties of this type of material. This study is a preliminary approach to the analysis of materials before their clinical testing.

## 2. Materials and Methods

### 2.1. Materials

The composition of the experimental infiltrants used for this study can be seen in Table 1. A series of six mixtures of monomers were prepared in different combinations and proportions. Bis-GMA_336_ and Bis-GMA_335_ were obtained from the Babeș-Bolyai University, “Raluca Ripan” Institute for Research in Chemistry (ICCRR, Cluj-Napoca, Romania) [11]. The remaining components were purchased from Sigma Aldrich Chemical Co. (Taufkirchen, Germany). As a photochemical initiation system, 0.5% camphorquinone (CQ) and 1% dimethylaminoethyl-methacrylate (DMAEM) were incorporated into each monomer mixture. Each mixture of monomers was divided into two parts, so that part of each composition made up the series of infiltrants P1–P6. The second series of the infiltrants (P1F–P6F) contained the same proportion of silanized BaF_2_-based glass [12], obtained at ICCRR, added to each of the remaining monomer mixtures. The glass was introduced into the mixture of monomers with the role of providing, especially in the first few days, fluoride ions with the role of obtaining an antibacterial effect after the application of the infiltrants.

### 2.2. Degree of Conversion (DC)

The measurement for the degree of conversion was performed by Fourier-transform infrared spectroscopy with attenuated total reflection (FTIR-ATR), (Jasco Europe s.r.l. Cremella, Italy). Infrared spectra of the mixtures of methacrylic monomers and corresponding copolymers were recorded on a Jasco FTIR 610 spectrophotometer, in the range 4000–550 cm^−1^, using an attenuated total reflection (ATR) device with a horizontal ZnSe crystal (Jasco PRO400S). The resolution of the spectra was 4 cm^−1^, and scans were repeated 100 times. The spectra were recorded after the samples were placed in direct contact with ZnSe crystal. Spectral recordings were conducted 24 h after the polymerization (shape 4 mm × 0.5 mm) performed with an LED (Guilin Woodpecker Medical Instruments Co., Guilin, China) dental light-curing lamp for 20 s on each side. The corresponding baseline was drawn, and the absorbance value of the baseline was subtracted from the maximum absorbance value at the corresponding wavenumber. The percentage of reacted double bonds (double-bond conversion) was calculated using the following equation:C% = {1 − [Amet/Aarom] copolymer/[Amet/Aarom] monomer} × 100,(1)
where A represents the absorbance intensity for the C=C double bond (Ameth) in the monomer and polymer or the C–C bond (Aarom) in the aromatic ring.

In order to determine the quantity of unreacted methacrylate groups, the absorption band from 1637.27 cm^−1^ was used due to the valence vibrations of the C=C double bonds. As an internal standard, the absorption band of the phenyl group (Aarom) from 1608.34 cm^−1^ (samples P2, P6, P2F, P6F, and Icon) was used. Furthermore, other absorption bands from 1531.2 cm^−1^ corresponding to the valence vibrations of the NH bonds (samples P1, P3, P4 and P1F, P3F, P4F) and from 1714.41 cm^−1^ corresponding to the vibrations of valence of C=O bonds (sample P5) were used as standards intern. The ratios of the absorbances intensities of the bands C=C/C–C, C=C/N–H, and C=C/C=O were compared before and after polymerization.

### 2.3. Water Sorption (WS)

The WS was performed according to ISO 4049 [13]. Six disc-shaped specimens were prepared, with 15 mm diameter and 1 mm thickness, from each investigated material. The specimens were polymerized with an LED dental light-curing lamp (Guilin Woodpecker Medical Instruments Co., Guilin, China) for a period of 20 s at five different points on the surface of the specimen. After preparing the specimens, they were placed in a desiccator for 24 h. The specimens were weighed several times to constant weight. The diameter and thickness of each sample were measured using a three-point digital measuring device. Knowing the mean radius (r) and the mean thickness (h), the volume of each specimen (V) was calculated using the following equation:V = πr^2^h (mm^3^).(2)

The specimens were immersed in 50 cm^3^ of distilled water in individual polyethylene bottles, for 7 days, in a thermostated water bath at 37 °C (±2). After this period, the specimens were removed using tweezers, before being lightly dried on a piece of filter paper and then in the air for 15 s. Each sample was then weighed three times in succession, taking the average weight. The samples were weighed in chronological order by the same operator, using the same balance. The weighed mass was noted as m_2_. Afterward, the samples were stored in a desiccator device until a constant mass of dry samples, which was noted as m_3_. The water sorption values of the samples were calculated using the following equation:WS = (m_2_ − m_3_)/V (µg/mm^3^),(3)
where m_2_ is the mass of the sample after immersion, m_3_ is the final dry mass (after drying in the desiccator), and V is the volume of samples.

### 2.4. Scanning Electron Microscopy (SEM)

The structures of the sample surface obtained from the infiltrant with the highest degree of conversion and of the Icon sample, before storage in distilled water and then after storage for a 7 day period, were analyzed using a scanning electron microscope (SEM-Inspect S, FEI Company, Hillsboro, OR, USA) at 2000× magnification.

### 2.5. Fluoride Release

To determine the amount of fluoride ions released, five disc-shaped specimens were prepared, having a diameter of 15 mm and a thickness of 1 mm, from each infiltrant investigated. The specimens were placed in Teflon molds and polymerized by exposing them to visible radiation generated by an LED dental light-curing lamp (Guilin Woodpecker Medical Instruments Co., Guangxi, China), for 20 s, at five points, on both sides of each sample. The specimens were then immersed in a solution containing 45 cm^3^ of double distilled water and 5 mL of total ionic strength adjustment buffer (TISAB III, HI 4010-06, Hanna Instruments, Woonsocket, RI, USA) at a constant temperature of 37 °C, performing daily measurements for 3 days, and then at 7 days. After each measurement, each sample was placed in a polyethylene container and stored at 37 °C in a thermostated water bath. A selective fluoride ion electrode was used for the analysis of fluoride ions released (Hanna instruments, Woonsocket, RI, USA). As a reference electrode, a combination fluoride electrode HI 4110 filled with HI 7075 electrolyte was used. In order to pre-calibrate the electrode, a series of calibration solutions was used, with different concentrations ranging from 10^−5^ to 10^−1^ mol/L, starting with a basic solution consisting of 1 M NaF (Merck). The calibration solutions were used in order to mark the calibration curve. All measurements, for both the investigated and the calibration solutions, were performed in 50 cm^3^ of a solution containing double-distilled water and TISAB III buffer (45/5) at a temperature of 37 °C (±2). The fluoride ion release was expressed in ppm.

### 2.6. Flexural Strength and Young’s Modulus

The flexural strength determination was recorded according to ISO 4049/2019 [13]. For each material presented in Table 1, five parallelepiped-shaped specimens were made using a Teflon mold (25 mm × 2 mm × 2 mm). The photopolymerization was performed with an LED dental light-curing lamp (Guilin Woodpecker Medical Instruments Co., Guilin, China), for 20 s, at five points of the sample, on both sides. After that, the cured specimens were stored in distilled water at a temperature of 37 °C, for 2 h. The three-point test was recorded according to ISO 4049/2019 on a mechanical testing machine (LF Plus, LLOYD, Instrument, Ametek Inc., West Sussex, UK), with cell movement speed = 0.5 mm/min and cell load = 50 N. The modulus of elasticity was calculated using NEXYGEN software coupled to the universal testing machine, which recorded the continuous response of the material to the force applied throughout the test.

### 2.7. Residual Monomers

*Sample preparation for HPLC analysis*. After a period of 7 days, the storage medium, consisting of distilled water where the specimens were immersed for the sorption test, was frozen. It was then lyophilized until the complete removal of the liquid, using a Model Alpha 1–4 LDPLUS lyophilizer. Subsequently, the residue containing the extracted residual monomers was resuspended in 2 mL of acetonitrile, and then filtered through 0.45 μm PTFE filters; finally, HPLC was used for the analysis.

*Instrumentation and method*. The analysis was performed using a Jasco HPLC chromatograph (Japan) equipped with a UV/Vis detector (UV-975), an HPLC pump (PU-980), a column thermostat (CO-2060 Plus), a ternary gradient unit (LG-980-02), and an injection valve equipped with a 20 μL sample loop (Rheodyne). Using a Hamilton Rheodyne syringe (50 mL), samples were injected manually. ChromPass software was used to control the system and analyze the experimental data. The separation was performed using a Lichrosorb RP-C18 column (25 cm × 0.46 cm), while the column temperature was set to 21 °C. The mobile phase consisted of a mixture of water (Millipore ultrapure water) and acetonitrile (A, HPLC grade). A gradient was applied as follows: 0–15 min, linear gradient 50–80% A; 15–25 min, linear gradient 80–50% A. The injector volume was 20 μL, while the flow rate was 0.9 mL·min^−1^. Because all the analytes (BisGMA, TEGDMA, HEMA, and UDMA) showed a significant absorption at 204 nm wavelength, UV detection was performed at this wavelength to monitor their elution. Samples were injected manually with a Hamilton Rheodyne syringe (50 mL). To control and analyze the experimental data, ChromPass software was used.

### 2.8. Statistical Analysis

The data were subjected to ANOVA and the Tukey test for post hoc comparisons between of the experimental infiltrants with filler and Icon, and the significance level was set to *p* = 0.05, using the Origin2019b Graphing and Analysis software.

For each group, 10 specimens were made, after which we were able to build a confidence interval to visually estimate the population parameter with a declared confidence level from a normal distribution using histograms. Deviations between media model predictions and observed data which exceeded ±15% were eliminated (choosing to list only six samples for each group) to obtain valid statistical inferences, such as confidence intervals and *p*-values.

## 3. Results

### 3.1. Degree of Conversion (DC)

In Figure 1, the same-colored bars represent the infiltrants with the same polymeric matrix. The difference between P and PF series samples is that the PF samples contained 5% fluoride releaser glass filler. From the graph, it can be seen that, at 24 h after polymerization, the experimental infiltrants containing Bis-GMA (P2 and P2F) and Bis-GMA F (P6 and P6F) had a degree of conversion over 80%. In this case, it can be observed that the degree of conversion increased slightly with the addition of a small amount of filler. The conversion also increased slightly with the addition of filler in the case of infiltrants with only TEGDMA/HEMA dilution monomers in the polymer matrix (P5 and P5F). In the case of infiltrants containing UDMA (P1, P3, P4 and P1F, P3F, P4F) a slight decrease in the degree of conversion was observed with addition of the filler. In the group of infiltrants (P1–P6) that did not contain the filler, the highest degree of conversion was recorded in the case of P6 infiltrant, with Bis-GMA F (89.70%) and the lowest (58.27%) was recorded in the case of P5 infiltrant, with only dilution monomers in the resin mixture. For infiltrants with filler added, the highest conversion (89.99%) was recorded for the P6F infiltrant with Bis-GMA_335_ and the lowest (60.62%) was recorded in the case of the P3F infiltrant (with UDMA/TEGDMA). For Icon, the commercial infiltrant, the degree of conversion was 46.94%.

### 3.2. Water Sorption

Figure 2 shows that the highest amount of water absorbed (33.3 µg/mm^3^) was recorded in the case of the P5F (TEGDMA/HEMA) infiltrant which can be explained by the hydrophilic character of the HEMA monomer. The lowest amount of water absorbed (5.42 µg/mm^3^) was recorded in the case of the P3F (TEGDMA/UDMA) infiltrant, most likely due to the hydrophobic nature of TEGDMA. In the case of P2F ad P6F infiltrants, which contained Bis-GMA in the polymer matrix, a lower sorption (16–17 µg/mm^3^) was recorded. The P4F infiltrant (with Et-OH added) absorbed a water amount of 8.53 µg/mm^3^, which slightly higher than for the P3F infiltrant (with quite similar composition). For the P1F (TEGDMA/HEMA/UDMA) infiltrant, the recorded sorption value was 14.48 µg/mm^3^, close to that of the Bis-GMA infiltrants in the composition. A sorption value of around 11 µg/mm^3^ was recorded for Icon.

Statistically, it was concluded that there are significant differences between the groups analyzed for sorption (*p* = 9.00128 × 10^−11^). Following the Tukey absorption test, the differences between the values were evident; P5F was different to all other samples (P1F–P6F), whereas P3F, P4F, and P5F samples were statistically different from the P1F, P2F, and P6F samples. Icon showed not statistically different values compared to P1F and P4F.

### 3.3. Microstructural Analysis

Figure 3 shows the SEM images (×2000) of the surfaces of the P2F and Icon samples (a,b) before and (c,d) after immersion in water. The P6F infiltrant had the highest conversion rate.

In both cases, after 7 days of immersion in water, obvious changes in the microstructure of the surfaces of the investigated samples were observed. There was a degradation of the sample surface, more accentuated in the case of the Icon sample, whose majority composition was represented by TEGDMA.

### 3.4. Fluoride Release

In Figure 4, it can be seen that the highest cumulative amount of fluoride released (26.94 ppm) was reached on day 7 in the case of the P5F (with only dilution monomers in composition) infiltrant. In fact, throughout the investigation period, the highest values of fluoride released were recorded in the case of the P5F infiltrant. At the opposite pole, the P3F (UDMA/TEGDMA) infiltrant was found to release the lowest amount of fluoride (2.15 ppm cumulative amount of fluoride, released on day 7). It can be noted that, in the case of the group of infiltrants with Bis-GMA (P2F and P6F), the cumulative amount of fluoride ions, released on the seventh day, was 8.66 ppm and 9.18 ppm, respectively, similar to that recorded in the case of infiltrant with P1F (TEGDMA/HEMA/UDMA), also 8.66 ppm. The P4F infiltrant had a similar composition to P3F (TEGDMA/UDMA) infiltrant, except that the percentage of TEGDMA was lower in the case of the P4F, which also had Et-OH. Therefore, the cumulative amount of fluoride released, on the seventh day, was 5.30 ppm for the P4F infiltrant. It is, therefore, supposed that the addition of ethyl alcohol favored obtaining a surface microstructure capable of favoring the fluoride release. It should also be remembered that the filler was added in a small quantity (5%) to the experimental infiltrants.

Analyzing the statistical results, ANOVA revealed that the values obtained regarding the evolution of the amount of fluoride for each of the six samples over the 7 day interval showed small significant differences (*p* = 3.27834 × 10^−4^). The Tukey test showed that the only differences were present between the P5F sample and the remaining samples.

### 3.5. Flexural Strength and Young’s Modulus

Following the mechanical test (Figure 5), in the case of P2 and P6 infiltrants (TEGDMA/HEMA/Bis-GMA), flexural strengths of 133.94 ± 16.389 MPa and 146.31 ± 7.032 MPa, respectively, were recorded. Despite being added in small quantities (5%), the inorganic filler contributed to a decrease in resistance to 94.337 ± 10.561 MPa for the P2F infiltrant and 94.083 ± 9.790 MPa for the P6F infiltrant. The lowest flexural strength (68.873 ± 5.351 MPa) was recorded for the P4 infiltrant, which was composed of TEGDMA/UDMA/Et-OH, followed by P5 infiltrant (122.85 ± 1.588 MPa) with a composition based only on dilution monomers. In the case of infiltrants with a small addition of filler, it was noted, as in the previous cases, a decrease in the flexural strength: 48.292 ± 3.095 MPa in the case of P4F and 67.864 ± 4.308 MPa in the case of P5F. In the case of the P3 infiltrant, the highest flexural strength (152.03 ± 8.157 MPa) was recorded; the absence of Et-OH in its composition compared to the P4 infiltrant led to a higher flexural strength. Furthermore, the absence of HEMA in the composition of the P3 infiltrant, compared to the P1 infiltrant, also led to a higher strength. In this case, the flexural strength of P3F infiltrant also decreased (101.86 ± 8.029 MPa) due to the addition of a minimum amount of filler. In the case of the P1 infiltrant (TEGDMA/HEMA/UDMA), the flexural strength was 138.98 ± 9.064 MPa, close to that of the infiltrants with Bis-GMA in their composition instead of UDMA. A lower flexural strength (58.8 ± 4.5 MPa) was recorded for the Icon infiltrant. It can be said that the mechanical properties may depend on the structure of the polymeric matrix, the hydrophilic or hydrophobic character of the monomers in the resin mixture, and the presence or absence of inorganic filler. In this study, all infiltrates with added filler (P1F–P6F) showed lower flexural strengths than their counterparts (P1–P6) without filler in the polymer matrix.

Statistically, it was concluded that there were significant differences between the analyzed groups (*p* = 2.64393 × 10^−4^). Following the Tukey test, the differences between the infiltrants without filler in their composition compared to those with filler were evident. There were no statistically significant differences between the infiltrants with filler and Icon.

The classification of experimental infiltrants, according to Young’s modulus (Figure 6), was as follows: (a) for those without filler in their composition, P6 > P2 > P3 > P1 > P5 > P4; (b) for those with filler in their composition, P3F > P1F > P2F > P5F > P6F > P4F.

According to the order in (a), the highest values for YM were recorded for infiltrants with Bis-GMA (as base monomer) in their composition. For the P6 infiltrant, an average value of 6.831 ± 0.148 GPa was obtained, slightly higher than that for the P2 infiltrant. In the case of infiltrants P1 and P3 (with UDMA in their composition), the values recorded for YM were comparable, around 4 GPa, lower in the case of P4 (3.054 ± 0.571 GPa), due to the addition of Et-OH which caused a reduction in the recorded values. In the case of P5 (TEGDMA/HEMA), the YM value was 3.423 ± 0.427 GPa.

According to the order in (b), it can be noticed that, although a small amount of filler was added, it had a very different influence on each type of resin composition. The highest value of YM, 5.127 ± 0.738 GPa, was recorded in the case of the P3F infiltrant (TEGDMA/UDMA), followed by that recorded in the case of the P1F infiltrant (TEGDMA/HEMA/UDMA) and the P2F infiltrant (TEGDMA/HEMA/Bis-GMA). The lowest value of YM (1.957 ± 0.871 GPa) was recorded, as in the order in (a), for P4F (TEGDMA/UDMA/Et-OH). Considering that, at the two poles of the order in (b), there were two infiltrants with a quite similar composition (TEGDMA/UDMA), it seems that the addition of Et-OH contributed to a significant decrease in YM. In the case of the P5F (TEGDMA/HEMA) and P6F (TEGDMA/HEMA/Bis-GMA_335_) infiltrants, the YM values were similar, over 2.5 GPa.

For Icon, the lowest values of the modulus of elasticity were registered: 1.3 ± 0.2 GPa. The incorporation of silanized glass nanoparticles into the experimental resins influenced the modulus of elasticity of these materials differently.

Analyzing the statistical results, ANOVA indicated that there was a major difference between each sample of experimental infiltrants (with or without powder) and Icon according to the Young’s modulus (*p* = 2.15837 × 10^−22^). The Tukey test showed very few statistically representative values, without differences between groups P1F–P3, P2F–P4, P5, P3F–P2, P5F–P4, and P2F. The Young’s modulus values for the Icon infiltrant presented different statistics compared to all experimental infiltrants.

### 3.6. Residual Monomers

The storage solutions of Bis-GMA, TEGDMA, HEMA, and UDMA reference standards (1 mg·mL^−1^) were prepared in acetonitrile and stored at 4 °C. The linearity of the response to the analytes was established with four concentration levels, and the regression factors *R*^2^ were 0.998. The presence of the aliphatic monomer TEGDMA was detected in all investigated samples.

Figure 7 shows the HPLC chromatograms of the standards of studied monomers and of the experimental infiltrant (P2F).

From Figure 8, it can be seen that the highest amount of TEGDMA (364.433ppm) was extracted from the experimental P5F infiltrant (with TEGDMA/HEMA). Dilution monomers being lighter and more flexible tended to be released first, depending on the quality of the polymerization. A similar value (360.92 ppm of extracted TEGDMA), detected by HPLC analysis, was recorded for the P6F experimental infiltrant (TEGDMA/HEMA/Bis-GMA F), followed by Icon (248.810 ppm of extracted TEGDMA). The lowest amount of TEGDMA (37.840 ppm) was extracted from the experimental P3F infiltrant (TEGDMA/UDMA). Close values of extracted TEGDMA, around 52 ppm, were recorded for the P1F (TEGDMA/HEMA/UDMA) and P4F (TEGDMA/UDMA/Et-OH) infiltrants. Thus, in the case of infiltrants with UDMA in the composition, the amount of TEGDMA extracted was lower.

Regarding UDMA, the highest amount (94.959 ppm) was extracted from the experimental infiltrant P4F (TEGDMA/UDMA/Et-OH), followed by P1F (52.244 ppm) and P3F (37.584 ppm). The highest amount of HEMA (199.294 ppm) was extracted from the P6F infiltrant (TEGDMA/HEMA/Bis-GMA F), followed by the P5F (87.18 ppm) and P1F (13.967 ppm) infiltrants. HEMA was not detected in P2F (TEGDMA/HEMA/Bis-GMA), but 7.938 ppm of Bis-GMA was extracted from this infiltrant. The presence of the characteristic Bis-GMA-F peak was not detected by HPLC analysis.

The classification of the total quantities of extracted residual monomer was as follows: P6F > P5F > Icon > P4F > P1F > P2F > P3F. The lowest value (75.4 ppm) was detected for the P3F infiltrant, and the highest value (560 ppm) was detected for the P6F infiltrant; residual TEGDMA (248 ppm) was detected for Icon.

## 4. Discussion

The polymeric matrix of the dental composites may contain different ratios of monomers with different viscosities. From the literature, it can be seen that the degree of conversion of the composites could increase or decrease depending on the ratio between the base monomer and dilution monomer. Bis-GMA could provide a conversion rate of up to 34.5–39%, whereas TEGDMA could provide a conversion rate of 75.5–82.5%, and UDMA could provide a conversion rate of 69.6–72.4% [14,15].

Bis-GMA, having a higher weight, is more rigid and reaches the gel state faster, while retaining unreacted double bonds. Adding TEGDMA, which is lighter and more flexible (reaching the gel state in a longer time), offers the possibility to react with as many Bis-GMA double bonds as possible, prolonging the time until the gel state appears. Therefore, a more fluid polymeric matrix yields a higher conversion. However, it has been found that the addition of a higher percentage of dilution monomer may increase the polymerization shrinkage. The addition of UDMA, which like TEGDMA has a linear chain with two aliphatic urethane bonds, can lead to the formation of hydrogen bonds, favoring an increase in the conversion degree and, thus, an improvement of the mechanical properties [16,17].

Unlike the Bis-GMA/TEGDMA-based mixtures, where the absorption standard of the phenyl group (Aarom) from 1608.34 cm^−1^ was used as an internal standard, in the case of the other mixtures in which the monomers did not contain aromatic structures, according to the literature data, the band from 1715–1720 cm^−1^ of the tensile vibrations of the C=O bond was used as an internal standard. In this case, however, underestimated conversion values were obtained [18,19]. In Guerra’s study, the absorption band from 1537 cm^−1^ was used as an internal standard for UDMA-based mixtures [20].

The following parameters had a significant role for obtaining an efficient polymerization and a high degree of conversion: type of initiation system and concentration chosen for initiators, curing time, sample thickness, filler content, and source of polymerization. It was also shown that the degree of conversion of the completely aliphatic polymers is higher than that of the polymers with aromatic and cycloaliphatic moieties, and that a longer chain leads to a higher degree of conversion. In the literature, for a Bis-GMA/TEGDMA 10/90% mixture, a degree of conversion of 76% was determined [21].

The difference in the values of the degree of conversion in the case of samples without Bis-GMA compared to the literature data may be due to the internal standards used for determination [22].

The conversion values previously reported for resins varied between 55.0% and 75.0%, and the results of a previous study showed for the Icon infiltrant resin a high DC of 80.9%, obtained after 40 s of light curing [23]. In another study, however, Gaglianone et al. [24] reported a DC (at 25 °C) for Icon of 41.3% (±1.1%).

In the present study, for the experimental infiltrants, in the samples without filler, a DC between 58.27% and 89.70% was obtained. For the Icon infiltrant, a DC of 46.94% was obtained. In case of the infiltrants with filler in the composition, the DC was in the range of 60.62–89.99% 24 h after polymerization.

The presence of TEGDMA in organic matrix of the resin infiltrants has several major disadvantages such as high water absorption, low mechanical properties, and low color stability. If the resin matrix also contains Bis-GMA, very stable acrylic bonds are formed along with the silanized filler particles, leading to the formation of a three-dimensional network that offers good mechanical and chemical characteristics [25].

Water sorption is dependent on the chemical structure of dimethacrylate polymer networks. A certain swelling capacity can be benefic in dental applications because it can compensate for the deformations that may occur as a result of polymerization shrinkage. The theoretical crosslinking density and the chemical structure of the monomer provide elasticity and hydrophilicity to the polymer chain, thereby influencing the water sorption. TEGDMA can be physically crosslinked due to the acceptor oxygen atom of the ester and ether groups. This crosslinking is possible only if TEGDMA is copolymerized with another dimethacrylate, which can provide a proton donor. The structure of the polymer network can then shrink due to the physical crosslinking, resulting from the hydrogen bonds, and this may limit the swelling of the material with water [26]

There are two theories regarding the diffusion of water into a resin-based material: (1) the free volume theory, according to which the water simply diffuses into the polymer network through cracks and imperfections in the network (air-filled voids) or at the resin–filler interface; (2) the theory of interaction, according to which the water molecules bind to ionic groups in the polymer chain through hydrogen bonds. Regarding the resin matrix, UDMA is more flexible compared to Bis-GMA, being less hydrophilic than it and TEGDMA. However, the presence of UDMA can improve DC, due to the higher degree of crosslinking compared to Bis-GMA. Therefore, the mixtures with Bis-GMA are more predisposed to sorption than those with UDMA [27]. In Figure 2, it can be seen that from the series of the experimental infiltrants with filler that the lowest values of WS were obtained in the case of P1F, P3F, and P4F materials with UDMA in the composition of the resin matrix.

According to ISO standard 4049:2009, WS values below the limit of 40 mg/mm^3^ are considered acceptable [13]. In our study, WS values for all the investigated infiltrants were below this limit.

Compared to Icon, which has a TEGDMA-based matrix, prone to swelling due to its hydrophilic nature, P6F had in the composition of the resin matrix, in addition to TEGDMA, HEMA, and Bis-GMA, a small percentage of inorganic filler. Although a higher DC than Icon was recorded for the P6F sample, the absorption was higher for the P6F sample. Examining the SEM images of the two sample surfaces, before immersion in water, it was observed that P6F had the smoothest surface. Small ditches and pits could be seen on the initial surface of the Icon sample. After 7 days of immersion in water, a deterioration of the surfaces of both samples was observed. In the case of Icon, larger cracks were observed, as well as detachments of material from the surface. In the case of P6F, the filler particles on the surface were more evident, which means that, even in this case, by releasing fluoride, water could penetrate more easily, damaging the surface of the samples over the time.

Fluoride release from a material is achieved by diffusion in an absorbed aqueous medium. The materials must have the ability to support the diffusion of water, while a balance must be maintained in terms of water sorption, which can lead to a deterioration of the material structure and a decrease in its properties. The hydrophilic order of the most common monomers used in dental materials is as follows: TEGDMA > BisGMA > UDMA [28]. Unlike giomers, where fluoride comes from the continuous dissolution of the glass core of the pre-reacted glass particles (SPRG) using acidified water from the hydrogel layer surrounding that core, in the case of nano-hybrid resins, fluoride is released by a diffusion process [29].

In the case of the experimental infiltrants (P1F–P6F series), the same concentration and the same type of fluoride-based glass filler were used. The differences were not related to the fluoride content in glass. Therefore, the differences between the amounts of fluoride ions released were related to the way in which they were released. In this case, the release of fluoride depended on the permeability of the polymer matrix, i.e., on how water helped to diffuse ions in the storage medium. Figure 2 and Figure 3 confirmed the correlation between water sorption and fluoride release. The amounts of fluoride released/day were quite small, due to the low concentration of added filler.

Gaglianone et al. reported in their study [24] the highest flexural strength and modulus of elasticity for a material composed of BisEMA (ethoxylated bisphenol A dimethacrylate)/TEGDMA/HEMA. Therefore, a composition based on monomers that can form crosslinks between chains can ensure a consolidation of the polymer network, thereby improving the general properties of the material. The flexural strength reported for this material (at 25 °C) was 103.2 MPa, and the modulus of elasticity was 2 GPa.

In the present study, for the P2 and P6 infiltrants based on Bis-GMA/TEGDMA/HEMA, flexural strengths of 133.94 ± 16.389 MPa and 146.31 ± 7.032 MPa, respectively, were recorded. The flexural strengths decreased below 100 MPa when a small amount of filler was added (P2F and P6F infiltrants). The lowest flexural strengths recorded (68.873 ± 5.3510 MPa/48.292 ± 3.095 MPa) were for the P4 and P4F experimental infiltrants which contained ethanol (TEGDMA/UDMA/Et-OH). The fact that ethanol negatively affected the mechanical properties, favoring the dilution of a polymer network under formation, was also confirmed in a previous study [30]. For a TEGDMA/Et-OH mixture (90:10) (at 25 °C), it was reported, in Gaglianone’s study, a flexural strength value of 37.2 ± 8.8 MPa [24]. For Icon (with TEGDMA in its composition), a flexural strength of 58.8 MPa, as in present study, and a modulus of elasticity of 1.3 GPa was reported.

The residual monomer release may influence the structural stability, degree of wear, and biocompatibility of the material. Furthermore, the following responsible factors for the residual monomer release can be mentioned: the degree of polymerization, the chemical nature, the size of the components released, and the chemistry of the solvent [31,32,33]. In the literature, there are reports concluding that monomers such as TEGDMA, with low molecular weight, can be extracted in larger quantities, being more mobile than high-molecular-weight monomers. Moreover, compared to UDMA, BisGMA, with a higher weight, is stiffer and will be harder to elute than UDMA [34]. In order to prevent the elution of large quantities of residual monomer from resin-based materials, a high degree of polymerization must be applied [35].

The limitation of this study is that only information related to the materials was obtained. Clinically, the material type, intraoral environment, mastication, and antagonistic structure may play an important role in the wear behavior of the material [36,37]. In the future, it will be important to study the wear behavior of the antagonist material and not only the infiltrant material, because the mechanical and chemical properties of the antagonist material are influenced by the wear that occurs between at least two opposite surfaces.

## 5. Conclusions

These in vitro tests were chosen because they are interdependent, they are relatively easy tests, and they can provide quick information that can be stored on a database for further investigation. Conversion to polymerization is a basic analysis to determine if the polymerization properly occurred. Depending on how the polymerization occurred, water absorption and the release of residual monomer and fluoride may have occurred. Mechanical properties are also influenced by conversion and absorption.

The DC differences in the case of infiltrants without Bis-GMA compared to the literature data may be due to the internal standards used. UDMA can improve DC, due to the higher degree of crosslinking compared to Bis-GMA, while the mixtures with Bis-GMA are more predisposed to sorption than those with UDMA. Therefore, the lowest values of WS were obtained in the case of P1F, P3F, and P4F materials with UDMA in the composition of their resin matrix. The highest amount of water absorbed was recorded in the case of the P5F (TEGDMA/HEMA) infiltrant, which can be explained by the hydrophilic character of the HEMA monomer. The amounts of fluoride released per day were quite small, due to the low concentration of added filler. From the SEM examination, it was observed that, after immersion in water, the sample surfaces were deteriorated, even for the sample with a higher DC.

It can be concluded that the mechanical properties may depend on the structure, the hydrophilic or hydrophobic character of the monomers in the resin mixture, and the presence or absence of the inorganic filler. In this study, all infiltrates with filler (P1F–P6F) showed lower flexural strengths than their counterparts (P1–P6) without filler in the polymer matrix. It seems that the addition of Et-OH contributed to a significant decrease in YM. Among the resin mixtures, the highest amount of eluted monomer was TEGDMA. Bis-GMA was found in the smallest amount. In the case of the (TEGDMA/UDMA/Et-OH) infiltrant, the amount of eluted UDMA was higher than that detected for the (TEGDMA/UDMA) infiltrant. It can be supposed that the presence of Et-OH favors the penetration of water into the network and, thus, the fluoride release and the release of the residual monomer. These results can be considered as a preliminary approach to the analysis of materials before their clinical testing.

## Figures and Tables

**Figure 1 materials-15-00803-f001:**
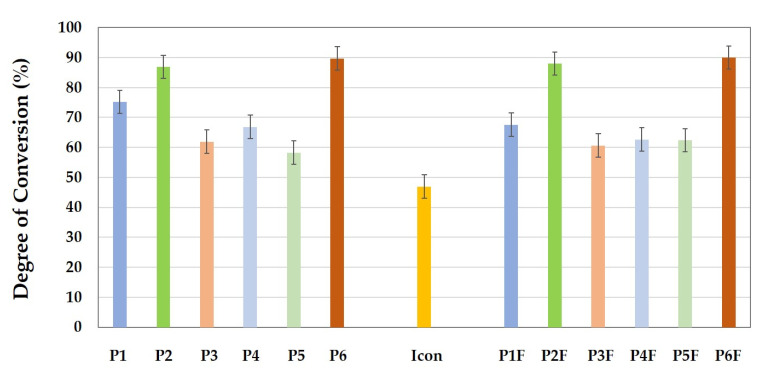
Graphical representation of the degree of conversion for the experimental infiltrants with and without filler in the composition, compared to the commercial Icon infiltrant, 24 h after polymerization.

**Figure 2 materials-15-00803-f002:**
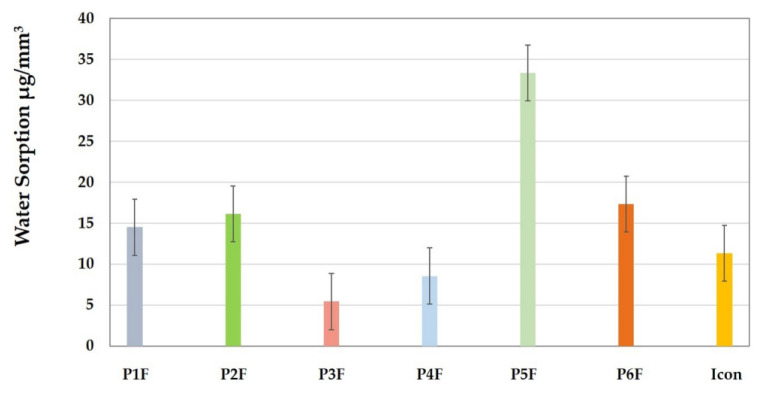
Sorption value of the investigated infiltrants (mean ± standard deviation).

**Figure 3 materials-15-00803-f003:**
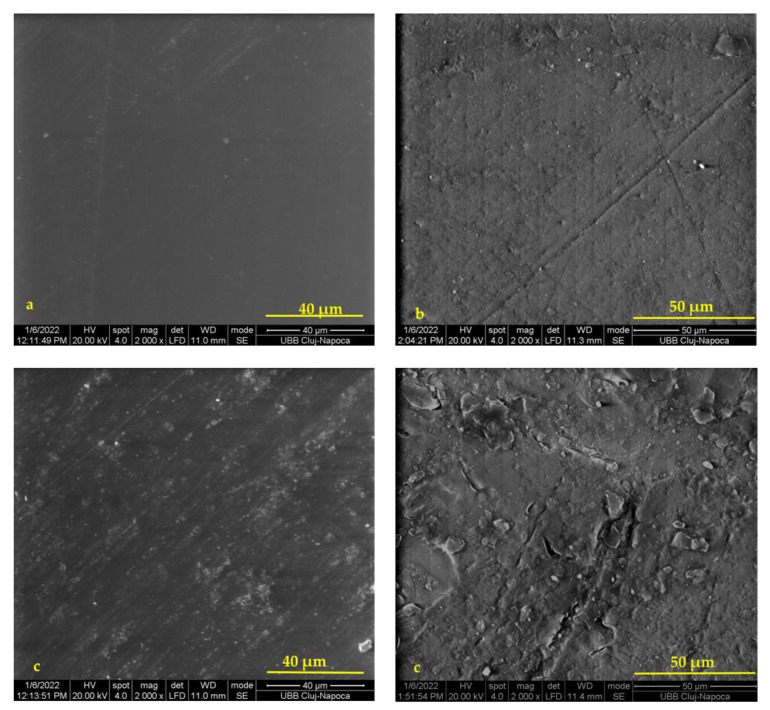
SEM images (×2000) of the surfaces of the P2F and Icon samples (**a**,**b**) before and (**c**,**d**) after immersion in water.

**Figure 4 materials-15-00803-f004:**
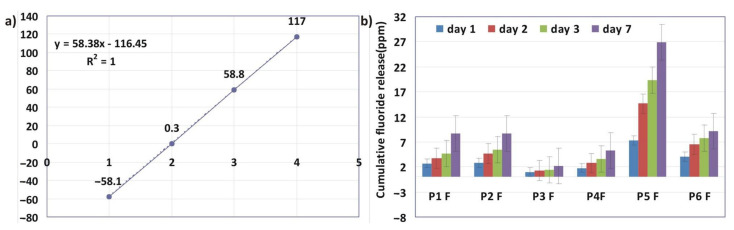
(**a**) The calibration curve of fluoride ions and (**b**) the cumulative fluoride release distribution over time for P1F–P6F experimental infiltrants.

**Figure 5 materials-15-00803-f005:**
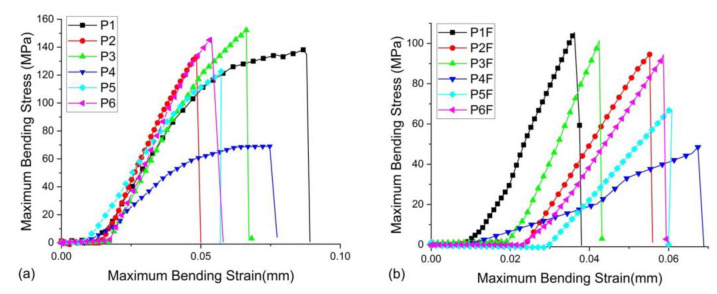
(**a**) Flexural strength of P1–P6 samples until fracture (**b**) Flexural strength of P1F–P6F samples until fracture.

**Figure 6 materials-15-00803-f006:**
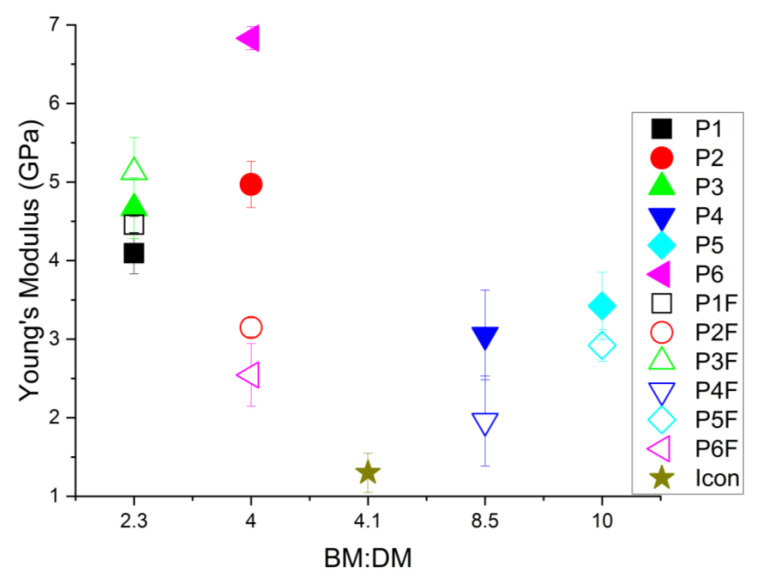
Graphical representation of Young’s modulus for experimental infiltrants (without/with filler in the resin mixture), as a function of the ratio between the base and the dilution monomer (BM:DM).

**Figure 7 materials-15-00803-f007:**
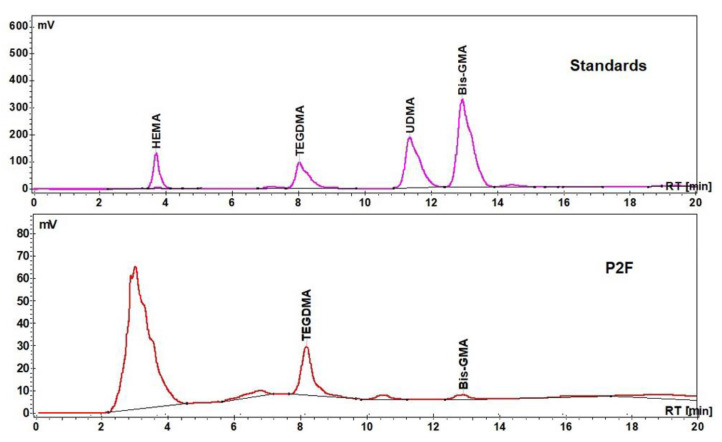
The HPLC chromatograms of standards (Bis-GMA, TEGDMA, HEMA, and UDMA) and an extract of the experimental infiltrant (P2F).

**Figure 8 materials-15-00803-f008:**
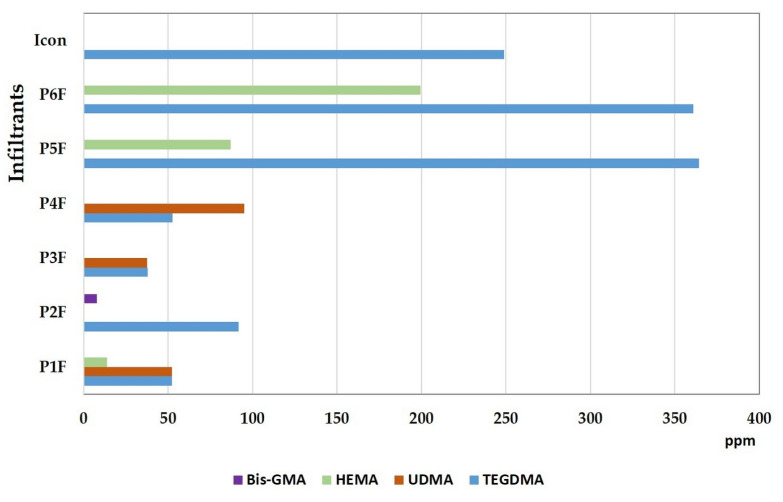
Quantitative values of the residual monomers for experimental infiltrants with filler in their composition, compared to Icon.

**Table 1 materials-15-00803-t001:** The composition of experimental resin infiltrants.

Infitrant Code/Monomers Series I	TEGDMA%	HEMA%	UDMA%	* BIS-GMA%	Et-OH%	Infiltrant Code/Monomer Series II	Resin Matrix%	*** Filler%
P1	50	20	30	-	-	P1 F	95	5
P2	60	20	-	20	-	P2 F		
P3	70	-	30	-	-	P3 F		
P4	55	-	30	-	15	P4 F		
P5	75	25	-	-	-	P5 F		
** P6	60	20	-	20	-	** P6 F		

* BisGMA_336_ analogue (93% 2,2-bis[*p*-(2-hydroxy-3-104 methacryloyloxypropoxy)-phenyl]-propane monomer and 7% dimer), synthesized in the ICCRR lab. ** P6 and P6F infiltrants contained BisGMA_335_, and the remaining infiltrants contained Bis-GMA_336_, synthesized in the ICCRR lab [11]. *** BaF_2_-based glass, synthesized in the ICCRR lab.

## Data Availability

Not applicable.

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
