# Peer review of "Development of New Experimental Dental Enamel Resin Infiltrants—Synthesis and Characterization"

_materials, 2022, doi:10.3390/ma15030803_

Round 1
Reviewer 1 Report
The article seems to be quite relevant in the field of biomaterial based on dental enamel resin. The authors analyzed and evaluated the changes in organic/inorganic structures of composite materials used as dental materials. Various analyzes of these structures have been carried out in many studies in the literature. However, the article can be made even better by making the following corrections. In this study authors say; The release of fluoride depends on the permeability of the polymer matrix, respectively by the 33 water sorption which may help to diffuse ions in the storage medium but which can, also, influence 34 the release of residual monomers. The highest flexural strengths were recorded for the 35 (TEGDMA/HEMA/Bis-GMA) infiltrants (133.94 ± 16.389 MPa/ 146.31 ± 7.032 MPa). The best alter- 36 native of the experimental infiltrants are P2 and P2F (Bis-GMA/HEMA/ TEGDMA). These results are scientifically important. However, the article can be made even better by making the following corrections
- The effect of the organic matrix structure (such as TEGDMA) on the mechanical and aesthetic behavior of the composite material should be emphasized.
for example; (authors can see)
“Investigation of two-body wear resistance of composite materials for biomaterial application in oral environment: the influence of antagonist materia”l and “Investigation of two-body wear resistance of composite materials for biomaterial application in oral environment: the influence of antagonist material”)
- A discussion section on water absorption in composite material should be developed. For example, how did the monomer structure of the organic structure affect this process and how could it affect the mechanical or aesthetic behavior of this material.

Author Response
Dear reviewer,
Thank you for your very helpful advices and guidelines regarding manuscript preparation. We have read your indications and we appreciate the suggestions. Supplementary information regarding the mechanical behavior and the water absorption were added. Also, supplementary data linked to the suggested references were added.

Reviewer 2 Report
The article is interesting. May interest the readers of the magazine. The microscopic images of the polymer before and after cross-linking are missing. The literature cited is quite old. There is no reference to new work on this topic. I suggest supplementing the literature review with new current items.
Author Response
Dear reviewer,
Thank you for your very helpful advices and guidelines regarding manuscript preparation. We have read your indications and we appreciate the suggestion regarding the microscopic images of the polymer before and after cross-linking
and more recent references inclusion. These aspects have been modified accordingly and added in the manuscript.
Reviewer 3 Report
The Authors studied two series of experimental infiltrants based on different monomers mixtures and without/with glass filler. The characteristics of the experimental infiltrants were investigated and the results were compared with those obtained for commercial Icon infiltrant designed to treat enamel white spots.
The "in vitro" study is aimed to evaluate two series of experimental infiltration materials, intended for the treatment of dental white spots and to investigate them in terms of the characteristics associated with this standard of material and, then, to compare them with the commercial infiltrant "Icon".
Summary:
The aim of the study has to be clearly declared
Introduction:
In lines 65 and 66 are reported considerations relating to carious cavities that are inappropriate since this study specifically deals with white spots (possibly of carious origin but not carious cavities), unless Authors used Icon proximal (inappropriate to treat white spots). The same consideration applies for the correspondent references (4 and 5).
M&M
The type of Icon used in comparison with the other systems has to be clearly declared: Icon proximal or Icon smooth surfaces. Of course, both the ICON versions are infiltrant.
Line 155 "after immersion (at a certain time)". At what timing after immersion? Was it the same timing for each of the specimens?
What was the power of the hypothesis test previously calculated and why did the Authors choose a parametric test with so few samples?
Discussion and conclusions
The discussion is well conducted in relation to results. However, conclusions fail to provide clear indications in relation to the study purpose that is to study infiltration materials"; for example, it is not easy to understand why "the best alternative of the experimental infiltrants are P2 and P2F". In the chapter of the conclusions the mechanical and chemical properties examined must be linked with what is required by the clinical application and not only taken up and discussed. The Authors have to explain clearly why they decided to choose these "in vitro" tests rather than others.
Therefore, before the conclusions it is necessary to introduce a "sub-chapter" declaring the limitations of this study. The Authors must consider that infiltrants are products designed to turn inside out the classic therapeutic approach of carious lesions. They have a deep conservative approach without requiring mechanical cavity preparation (no drilling). The infiltrant permeates damages produced in the enamel prisms acting as a hybridizer, supporting the residual prism structures and preventing corrosion processes, bacterial lysis, structural failures, etc.. In particular, for white spot treatment some mechanical characteristics are therefore absolutely secondary with respect to the infiltrating capacity. This study, although interesting, cannot determine it. However, it could suggest a preliminary flow chart to analyze infiltrant materials before clinical validation. This would be an interesting topic for the authors to discuss. It is inappropriate to jump directly to clinical conclusions based on this study
Author Response
Dear reviewer,
Thank you for your very helpful advices and guidelines regarding manuscript preparation. We have read your indications and the following modifications have been made to the manuscript:
Abstract
Point: The aim of the study has to be clearly declared
Reply: The aim was added
Introduction
Point: considerations relating to carious cavities that are inappropriate
Reply: they were removed and, also, the references related to them
Materials and Methods
Point: The type of Icon used in comparison with the other systems has to be clearly declared: Icon proximal or Icon smooth surfaces. Of course, both the ICON versions are infiltrant.
Reply: It was used Icon Smooth Surface. We added this detail in the manuscript
Point: Line 155 "after immersion (at a certain time)". At what timing after immersion? Was it the same timing for each of the specimens?
Reply: Weighing the samples was performed after 7 days, following the procedure described in the text; after buffering and air drying (15 seconds), the weighing of the samples was repeated 3 times in succession. The samples were weighed in chronological order by the same operator, using the same balance. The text between the brackets was removed in order not to create confusion and supplementary text was added in order to be more precisely.
Point: What was the power of the hypothesis test previously calculated and why did the Authors choose a parametric test with so few samples?
Reply: For each group, 10 specimens were made, after which we were able to build a confidence interval to estimate the population parameter with a declared confidence level from a normal distribution visually using histograms. Deviations between media model predictions and observed data, which exceed ± 15% deviations, were eliminated (choosing to list only 6 samples for each group) to obtain valid statistical inferences, such as confidence intervals and p-values. The details were added to the manuscript.
Discussion and conclusions
Reply: Up to this stage of the research, several variants of infiltrants were obtained, which were characterized in terms of quality of materials. Some of these results are presented in this study. The characterizations were performed, and compared with the data from the specialized literature. In the next stage, the materials will be tested in the clinic.
More details were added in the manuscript, in the discussion and conclusions chapters. We also added “Limitations of the study” section.
Round 2
Reviewer 3 Report
The Authors appropriately improved the quality of the study by specifying both the results and limits. It is fine for me.